# m6A Reader YTHDF2 Regulates LPS-Induced Inflammatory Response

**DOI:** 10.3390/ijms20061323

**Published:** 2019-03-15

**Authors:** Ruiqing Yu, Qimeng Li, Zhihui Feng, Luhui Cai, Qiong Xu

**Affiliations:** Guanghua School of Stomatology & Guangdong Provincial Key Laboratory of Stomatology, Sun Yat-sen University, Guangzhou 510055, China; yruiqing@163.com (R.Y.); liqimeng22@outlook.com (Q.L.); fzhh136@sina.com (Z.F.); lucy0526@126.com (L.C.)

**Keywords:** YTHDF2, inflammatory reaction, MAPK, NF-κB, mRNA stability

## Abstract

N6-methyladenosine (m6A) is an abundant mRNA modification that affects multiple biological processes, including those involved in the cell stress response and viral infection. YTH domain family 2 (YTHDF2) is an m6A-binding protein that affects the localization and stability of targeted mRNA. RNA-binding proteins (RBPs) can regulate the stability of inflammatory gene mRNA transcripts, thus participating in the regulation of inflammatory processes. As an RBP, the role of YTHDF2 in the LPS-induced inflammatory reaction has not been reported. To elucidate the function of YTHDF2 in the inflammatory response of macrophages, we first detected the expression level of YTHDF2 in RAW 264.7 cells, and found that it was upregulated after LPS stimulation. YTHDF2 knockdown significantly increased the LPS-induced IL-6, TNF-α, IL-1β, and IL-12 expression and the phosphorylation of p65, p38, and ERK1/2 in NF-κB and MAPK signaling. Moreover, the upregulated expression of TNF-α and IL-6 in cells with silenced YTHDF2 expression was downregulated by the NF-κB, p38, and ERK inhibitors. YTHDF2 depletion increased the expression and stability of MAP2K4 and MAP4K4 mRNAs. All of these results suggest that YTHDF2 knockdown increases mRNA expression levels of MAP2K4 and MAP4K4 via stabilizing the mRNA transcripts, which activate MAPK and NF-κB signaling pathways, which promote the expression of proinflammatory cytokines and aggravate the inflammatory response in LPS-stimulated RAW 264.7 cells.

## 1. Introduction

Inflammation has been viewed as an immune response to harmful stimuli, tissue injury, or infection that protects the body from various pathogens, such as bacteria, viruses, and fungi [1,2]. However, prolonged and excessive inflammation induces immune disorders and causes excessive tissue damage, resulting in many diseases, including arthritis, diabetes, cardiovascular disease, and cancer [3,4]. Macrophages are a vital component of the immune system, and play a large role in inflammatory processes. As important immune cells, macrophages respond immediately to microbial infection and directly engulf or indirectly eliminate pathogens by releasing multiple inflammatory mediators [5]. Lipopolysaccharide (LPS), a major component of gram-negative bacterial cell walls, is one of the most potent stimulators of innate immunity. LPS can stimulate macrophages to secrete various inflammatory cytokines, including TNF-α, IL-6, and IL-1β, by activating the NF-κB and MAPK pathways, further aggravating the immune response [6,7].

RNA-binding proteins (RBPs) interact with gene transcripts to regulate the maturation, transport, stability, and degradation of RNAs [8]. Some RBPs have been shown to participate in the modulation of immune and inflammatory processes. The RBP tristetraprolin (TTP) plays a significant role in regulation of the inflammatory response by promoting the degradation of proinflammatory cytokine mRNAs. TTP acts on proinflammatory cytokine mRNAs whose 3′UTRs have AU-rich elements (AREs), such as TNF-α, GM-CSF, IL-1β, and IL-6 mRNAs. TTP binds to AREs and recruits the CCR4-NOT1 deadenylation complex to the target mRNA, thereby promoting mRNA degradation [9,10,11]. Human antigen R (HuR) is another RBP that binds to the ARE of inflammatory cytokine mRNAs, but its role is opposite to that of TTP. HuR protects its target transcripts from TTP and directs them to ribosome complexes, enhancing the rate of translation [12].

N6-methyladenosine (m6A), methylation at the N6 of adenosine, has been a hot research topic recently, since the transcriptome-wide profiling of m6A was validated in 2012 [13,14]. More than 10,000 m6A peaks have been mapped and found to be on a consensus RNA motif, RRACH (R=A or G; H=A, U, or C) [15]. m6A is involved in diverse biological processes, including stem cell proliferation and differentiation, tumorigenesis, and viral infections [16,17,18,19,20], through m6A readers, writers and erasers by mediating RNA processing steps, such as splicing, translation and degradation [21,22,23,24]. The m6A reader YTHDF2 is a member of the YTH (YT521-B homology) domain family (YTHDF) proteins and preferentially binds m6A within a G(m6A)C consensus site [25,26,27]. YTHDF2 C-terminal YTH domain (YTHDF2-C) can selectively bind m6A, while the P/Q/N-rich N terminus can bring the target mRNA to cytoplasmic foci (P bodies) and recruit the CCR4-NOT deadenylase complex to the RNA [28,29]. Previous studies have shown that RBPs regulate the inflammatory response by affecting the stability of inflammatory gene transcripts. As an emerging RBP, YTHDF2 has been shown to regulate pathological processes by destabilizing mRNA [30,31]. We hypothesized that YTHDF2 might also play a role in regulation of the inflammatory reaction by destabilizing inflammation-related gene transcripts.

In this study, we aimed to investigate the potential function and molecular mechanism of YTHDF2 in the inflammatory reaction of mouse macrophages. YTHDF2 expression at both the mRNA and protein levels was detected in RAW 264.7 cells after LPS stimulation. Furthermore, YTHDF2 suppression was conducted to explore the regulatory role of YTHDF2 in LPS-induced proinflammatory cytokine expression, in addition to MAPK and NF-κB signaling activation. We also investigated the potential YTHDF2 target transcripts activating the MAPK and NF-κB pathways. Our study is the first to discover that YTHDF2 might play an important role in regulating inflammation and preventing an excessive inflammatory response.

## 2. Results

### 2.1. LPS Stimulation Increases YTHDF2 Expression in RAW 264.7 Cells

To explore the role of YTHDF2 in the LPS-stimulated inflammatory response of macrophages, we first analyzed YTHDF2 expression in RAW 264.7 cells stimulated with 1 μg/mL LPS for the indicated times (0 h, 3 h, 6 h, 12 h, and 24 h). The YTHDF2 mRNA level was significantly increased at 3 h and then gradually decreased to the normal level within 24 h (Figure 1A). The YTHDF2 protein level increased at 6 h and 12 h and then slightly decreased by 24 h (Figure 1B,C).

### 2.2. YTHDF2 Knockdown Promotes LPS-Induced Inflammatory Cytokine Expression in RAW 264.7 Cells

To further explore the effect of YTHDF2 on the LPS-induced inflammatory reaction in RAW 264.7 cells, the cells were transfected with siYTHDF2 (#1, #2, and #3) and NC to knock down YTHDF2 expression. The YTHDF2 mRNA and protein levels significantly decreased after gene knockdown (Figure 2A–C). siYTHDF2 #1 showed the highest knockdown efficiency and was used in the following experiments.

To investigate the regulatory role of YTHDF2 in LPS-induced inflammatory cytokine expression, siYTHDF2-treated RAW 264.7 cells were stimulated with 1 μg/mL LPS for the indicated times, and then mRNA levels of TNF-α, IL-6, IL-1β, and IL-12 were measured. Compared to the NC-treated group, the siYTHDF2-treated group showed significantly increased TNF-α and IL-6 mRNA levels after LPS stimulation at all the indicated time points within 24 h (Figure 2D,E). The IL-1β mRNA levels were upregulated at 12 h and 24 h (Figure 2F), while the IL-12 mRNA levels were upregulated at 6 h and 12 h (Figure 2G).

### 2.3. YTHDF2 Knockdown Has Little Effect on Cytokine mRNA Stability

To investigate whether YTHDF2 promotes the degradation of cytokine mRNA, an mRNA stability assay was conducted to measure the stability of TNF-α, IL-1β, IL-6, and IL-12 mRNAs. RAW 264.7 cells transfected with siYTHDF2 or NC were stimulated with 1 µg/mL LPS for 6 h and then treated with 5 µg/mL actinomycin D for the indicated times (0 h, 2 h, and 4 h). As shown in Figure 3, there were no significant differences in the mRNA stability of these cytokines between the siYTHDF2- and NC-treated groups.

### 2.4. YTHDF2 Knockdown Activates LPS-induced NF-κB and MAPK Signaling in RAW 264.7 Cells

Activation of the NF-κB and MAPK signaling pathways is required for LPS-stimulated proinflammatory cytokine expression in macrophages. To explore the function of YTHDF2 in NF-κB and MAPK signaling pathway activation, we measured the phosphorylation levels of p65, p38, ERK, and JNK by western blotting (Figure 4A–H). The results show that YTHDF2 depletion significantly upregulated p65, p38, and ERK phosphorylation but had little effect on JNK phosphorylation. Therefore, YTHDF2 knockdown activated p65 in the NF-κB signaling pathway and both p38 and ERK in the MAPK signaling pathway.

To confirm the role of NF-κB and MAPK signaling pathways in the expression of inflammatory cytokines in siYTHDF2-treated cells, RAW 264.7 cells were treated with the NF-κB inhibitor BAY 11-7082, the p38 inhibitor SB203580, the ERK inhibitor U0126, or the JNK inhibitor SP600125 to block signaling; then, the expression levels of TNF-α and IL-6 were evaluated. Our data show that the upregulated expression levels of TNF-α and IL-6 in siYTHDF2-treated cells were downregulated by the NF-κB, p38, and ERK inhibitors. The JNK inhibitor SP600125 did not significantly inhibit the expression of TNF-α or IL-6 (Figure 5). These results indicate that YTHDF2 suppression increased the expression of proinflammatory cytokines by activating the NF-κB, p38, and ERK signaling.

### 2.5. YTHDF2 Knockdown Increases MAP2K4 and MAP4K4 Expression and mRNA Stability

Components of the NF-κB and MAPK pathways can be phosphorylated by upstream phosphatases during the inflammatory response to LPS stimulation. To investigate the mechanism of the activating effect of YTHDF2 knockdown on LPS-induced NF-κB, p38, and ERK signaling, we examined upstream components of the NF-κB and MAPK pathways. MAP2K4 and MAP4K4 mRNA levels were measured in RAW 264.7 cells transfected with siYTHDF2 or NC after LPS stimulation. MAP2K4 and MAP4K4 mRNA expression levels were significantly increased in the siYTHDF2-treated cells (Figure 6A).

The mRNA stability assay was then performed to analyze MAP2K4 and MAP4K4 mRNA stability. As shown in Figure 6B,C, YTHDF2 depletion increased the stability of MAP2K4 and MAP4K4 mRNA transcripts. Overall, these results indicate that YTHDF2 knockdown upregulated the phosphorylation of NF-κB and MAPK signaling components through stabilizing MAP2K4 and MAP4K4 mRNAs, thus promoting the expression of proinflammatory cytokines in LPS-stimulated RAW 264.7 cells.

## 3. Discussion

m6A is the most widespread eukaryotic mRNA modification, and an increasing number of studies are focusing on its role in regulating various biological processes, such as those involved in hematopoiesis, neurodevelopment, and tumorigenesis [31,32,33,34]. Recent studies have demonstrated that m6A exerts a regulatory effect by interacting with m6A readers [28]. YTHDF is a family of cytoplasmic reader proteins that preferentially bind m6A within a G(m6A)C consensus site and regulate mRNA degradation and translation [21,25,26,27]. As an important m6A reader, YTHDF2 has been shown to promote RNA degradation and be involved in multiple physiological and pathological processes. YTHDF2 regulates the cell stress response through destabilizing HSP90, HSP60, and HSPB1 mRNAs, and reduces the relative viability of HepG2 cells [35]. The suppression of YTHDF2 promotes hematopoietic stem cell (HSC) expansion by increasing the stability of multiple mRNAs critical for HSC self-renewal [32]. YTHDF2-mediated notch1a mRNA decay contributes to the suppression of Notch signaling and thereby modulates HSC and progenitor cell specification during zebrafish embryogenesis [18]. These studies suggest that YTHDF2 plays its role by destabilizing the key gene transcripts in certain biological process. According to previous studies, the stability of the mRNA transcripts of immune- and inflammation-related genes can be regulated by some RBPs, such as TTP, HuR, and AUF1 [12,36]. Whether YTHDF2 plays a regulatory role in the inflammatory response by modulating mRNA stability is still unknown. To address this issue, our study examined YTHDF2 as a possible regulator of the inflammatory response in macrophages. RAW 264.7 cells have been widely used as a suitable macrophage model for research on inflammation. In the present study, the expression of YTHDF2 in LPS-stimulated RAW 264.7 cells was first detected. Increased YTHDF2 expression in response to LPS stimulation was found in the RAW 264.7 cells, suggesting that YTHDF2 might play a role in the inflammatory processes of macrophages. Our data showed that YTHDF2 mRNA increased in response to LPS during the first 6 h and then decreased. The expression pattern of YTHDF2 is similar to that of TTP. TTP plays an important role in the early stage of inflammatory response; its expression rapidly increases then decreases after LPS stimulation for 1 h [10]. We assumed that YTHDF2 might also participate in the regulation of the LPS-induced inflammation at an early stage in macrophages.

In response to the LPS stimulation, macrophages express proinflammatory cytokines, such as TNF-α, IL-1β, and IL-6, which can eventually have harmful consequences associated with the pathogenesis of inflammatory diseases [37]. To elucidate the regulatory role of YTHDF2 in the LPS-induced inflammatory response of macrophages, we used siRNA to knockdown YTHDF2 expression in RAW 264.7 cells and detected the accumulation of proinflammatory cytokines. The results show that YTHDF2 depletion led to increased TNF-α, IL-1β, IL-6, and IL-12 expression. To further verify whether YTHDF2 destabilizes gene transcripts of these inflammatory cytokines, we measured the stability of TNF-α, IL-1β, IL-6, and IL-12 mRNAs. Intriguingly, our results indicate that YTHDF2 had little effect on TNF-α, IL-1β, IL-6, and IL-12 mRNA stability. Therefore, we concluded that YTHDF2 might regulate the mRNA stability of upstream molecules in the LPS-induced inflammatory response of macrophages. 

LPS activates intracellular signaling cascades including the NF-κB and MAPK pathways, which further induce the expression of inflammatory mediators in immune cells [7]. According to a recent study, the overexpression of YTHDF2 suppresses the activation of MEK and ERK in hepatocellular carcinoma cells (HCC) [38]. To investigate the function of YTHDF2 during the activation of NF-κB and MAPK signaling, the present study examined the phosphorylation of several key molecules in those pathways by western blotting. The results show that the phosphorylation levels of p65, p38, and ERK were increased in siYTHDF2-treated RAW 264.7 cells. Furthermore, both NF-κB and MAPK signaling pathway inhibitors were used to confirm the roles of the two signaling pathways in upregulating cytokine expression. After the signaling pathways were blocked, the upregulated expression levels of TNF-α and IL-6 in siYTHDF2-treated cells were downregulated, confirming the roles of p65, p38, and ERK in upregulating the expression of cytokines after YTHDF2 suppression. The results indicate that YTHDF2 regulates inflammatory cytokine expression via the p65, p38, and ERK signaling pathways in macrophages.

MAP2K4 and MAP4K4 are upstream molecules in the LPS-induced inflammatory response and play crucial roles in MAPK and NF-κB signaling pathway activation. MAP2K4 is a member of the MAPKK family that can be phosphorylated by its upstream MAPKKKs to activate two downstream components of the MAPK signaling pathway (JNK and p38) [39,40]. It has also been reported to activate NF-κB signaling in response to cellular stress [41]. MAP4K4 can promote NF-κB localization and activation and activate ERK, p38, and JNK in MAPK signaling [42,43,44,45]. Recent studies have shown that MAP2K4 and MAP4K4 are the target genes of YTHDF2 [24,32]. Recent research has profiled YTHDF2 targets and binding sites by the combination of photoactivatable ribonucleotide crosslinking and immunoprecipitation (PAR-CLIP) and sequencing profiling of immunopurified ribonucleoprotein complex RNA (RIP-seq) in HeLa cells; the sequencing results show that human MAP2K4 mRNA is one of the target transcripts of human YTHDF2 [24]. Another study later determined YTHDF2 targets by performing infrared UV-crosslinking immunoprecipitation sequencing (irCLIP-seq) in mouse hematopoietic cells, and the sequencing data revealed mouse MAP4K4 as a target of mouse YTHDF2 [32]. To verify whether YTHDF2 affects the mRNA stability of MAP2K4 and MAP4K4 in LPS-stimulated RAW 264.7 cells, we measured the mRNA expression and stability of MAP2K4 and MAP4K4. The data showed that YTHDF2 knockdown increased the expression and stability of MAP2K4 and MAP4K4 mRNAs. Our findings indicate that YTHDF2 might be involved in regulation of the LPS-stimulated inflammatory reactions via regulating the stability of MAP2K4 and MAP4K4 mRNAs in RAW 264.7 cells.

The present study illustrates that YTHDF2 expression was increased in LPS-stimulated RAW 264.7 cells. Furthermore, YTHDF2 suppression increased the stability and enhanced the expression of MAP2K4 and MAP4K4 mRNAs, which consequently triggered the activation of p38, ERK, and NF-κB signaling, thus promoting the expression of TNF-α, IL-1β, IL-6, and IL-12. These results suggest that YTHDF2 might play a negative regulatory role in LPS-induced inflammatory responses of macrophages. The present study provides a potential target for anti-inflammatory therapies and new insight for further study of inflammatory mechanisms.

## 4. Materials and Methods

### 4.1. Cell Culture

RAW 264.7 murine macrophages were purchased from American Type Culture Collection (ATCC, Manassas, VA, USA). Cells were cultured in RPMI 1640 medium (Gibco, Carlsbad, CA, USA), supplemented with 10% fetal bovine serum (FBS; Gibco, Carlsbad, CA, USA), at 37 °C in a humidified atmosphere of 5% CO_2_ and 95% air. When the cells reached 80% confluence, they were harvested using cell scrapers and subcultured at a ratio of 1:3.

### 4.2. Cell Stimulation

RAW 264.7 cells were cultured in 6-well culture dishes until they reached approximately 80% confluence and were then stimulated with 1 μg/mL Escherichia coli LPS (InvivoGen, San Diego, CA, USA) for a certain number of hours. Cells not stimulated by LPS were used as a control.

In the signaling pathway inhibition experiments, the cells were first treated with the signaling pathway inhibitor BAY 11-7082 (Beyotime, Shanghai, China; 10 μM), U0126 (Beyotime, Shanghai, China; 10 μM), SB203580 (Beyotime, Shanghai, China; 20 μM) or SP600125 (Beyotime, Shanghai, China; 20 μM) for 1 h, and were then stimulated with 1 μg/mL LPS for 6 h. Cells not stimulated with LPS or treated with signaling pathway inhibitors were used as a blank control.

### 4.3. YTHDF2 Small Interfering RNA (siRNA) Transfection

RAW 264.7 cells were seeded in 6-well culture plates at a density of 5 × 10^5^/well and cultured for 24 h. The cells were then transfected with 50 nM siRNA (siYTHDF2 or the negative control, i.e., NC; Invitrogen, Carlsbad, CA, USA) and 7.5 μL of Lipofectamine^TM^ 3000 transfection reagent (Invitrogen, Carlsbad, CA, USA) per well for 24 h, according to the manufacturer’s instructions. The siYTHDF2 sequences are shown in Table 1.

### 4.4. Real-Time Quantitative Polymerase Chain Reaction (qRT-PCR)

Total RNA was extracted from RAW 264.7 cells using RNAzol (MRC, OH, USA) and purified according to the manufacturer’s instructions. One microgram of RNA was reverse-transcribed into cDNA using a PrimeScript^TM^ RT reagent kit (Takara, Kyoto, Japan). qRT-PCR was performed using a LightCycler 480 system with SYBR Green I Master Mix (Roche, Basel, Switzerland). The housekeeping gene GAPDH was used as the reference gene. The relative expression of the genes of interest was normalized to the geometric mean of GAPDH expression. Primer sequences are listed in Table 2.

### 4.5. Western Blotting

RAW 264.7 cells were harvested in protein lysis buffer including radioimmunoprecipitation assay (RIPA) lysis buffer (Beyotime, Haimen, China), a phosphatase inhibitor cocktail (Beyotime), and a protease inhibitor cocktail (Beyotime), and incubated on ice for 30 min. Bicinchoninic acid (BCA) protein assay (Beyotime) was used to measure the protein concentration. Forty micrograms of protein were separated by 10% sodium dodecyl sulfate–polyacrylamide gel electrophoresis (SDS-PAGE) and electrophoretically transferred onto a polyvinylidene fluoride (PVDF) membrane (Millipore, Billerica, MA, USA). Nonspecific protein binding was eliminated by blocking the membrane with 5% nonfat milk at room temperature for 1 h. The membrane was then incubated with primary antibodies against YTHDF2 (1:1000; Proteintech, Chicago, IL, USA), p65, p-p65, ERK, p-ERK, JNK, p-JNK, p38, p-p38, GAPDH, and vinculin (1:1000; Cell Signaling Technologies, Danvers, MA, USA), overnight at 4 °C. After washing, the membrane was incubated in HRP-conjugated secondary antibodies (1:1000; Cell Signaling Technologies, Danvers, MA, USA) at room temperature for 1 h. Antibody binding was developed using an enhanced chemiluminescence system (Millipore) and scanned by an ImageQuant LAS 4000 mini system (GE Healthcare Life Sciences, Chicago, IL, USA). The band densities were measured and normalized to that of GAPDH or vinculin using ImageJ v1.47 software (National Institutes of Health, Bethesda, MD, USA). ReBlot Plus (Millipore) was used to strip and reprobe the proteins when target proteins with similar molecular weights on the same membrane needed to be distinguished.

### 4.6. mRNA Stability Assay

The cells were stimulated with 1 μg/mL LPS for 6 h and then treated with 5 µg/mL actinomycin D (Merck) for 0, 2, and 4 h to inhibit global mRNA transcription. Extracted RNA was reverse-transcribed into cDNA. The mRNA transcript levels of the genes of interest were detected by qRT-PCR. mRNA stability was determined by the analysis of relative expression at 2 h and 4 h after actinomycin D treatment in RAW 264.7 cells.

### 4.7. Statistical Analysis

All results are expressed as the mean ± standard deviation (SD) of at least three independent experiments. Data were analyzed using SPSS v20.0 (SPSS, Inc., Chicago, IL, USA). The two-tailed unpaired t-test was used to compared two groups, and ANOVA was used for the comparison of multiple groups. The significance level was set at *p* < 0.05.

## Figures and Tables

**Figure 1 ijms-20-01323-f001:**
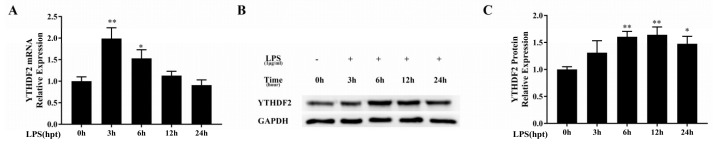
YTHDF2 expression in LPS-stimulated RAW 264.7 cells. RAW 264.7 cells were stimulated with 1 μg/mL LPS for 0 h, 3 h, 6 h, 12 h, and 24 h. (**A**) YTHDF2 mRNA expression level was quantified by qRT-PCR, and GAPDH was used as a normalization control; (**B**,**C**) YTHDF2 protein level was measured by western blotting, and GAPDH was used as an internal control. The results are shown as the mean ± SD (*n* = 3). The *p* values were calculated using one-way ANOVA. * *p* < 0.05, ** *p* < 0.01.

**Figure 2 ijms-20-01323-f002:**
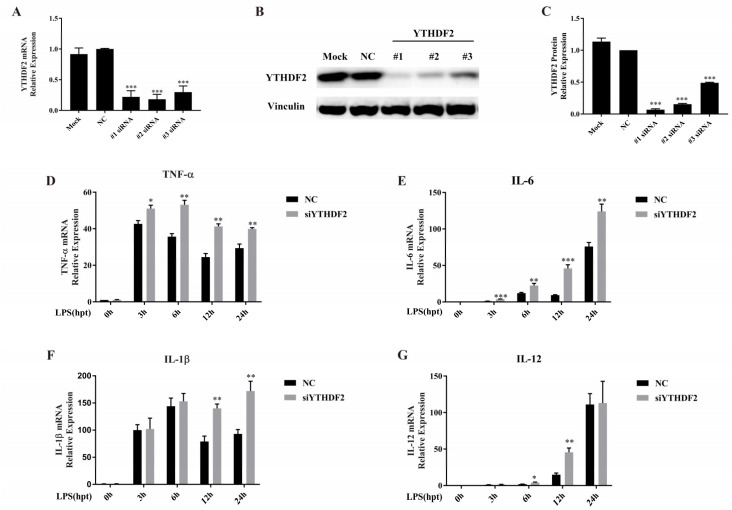
Effect of YTHDF2 knockdown on inflammatory cytokine expression in RAW 264.7 cells. (**A**–**C**) The transfection efficiency of YTHDF2 knockdown in RAW 264.7 cells was measured by both qRT-PCR and western blotting. Mock: cells treated with transfection reagent; NC: cells transfected with negative control siRNA; #n (*n* = 1, 2, 3) siRNA: cells transfected with YTHDF2 siRNA. The *p* values were calculated using one-way ANOVA; (**D**–**G**) RAW 264.7 cells were transfected with YTHDF2 siRNA (siYTHDF2) or negative control siRNA (NC) for 24 h and then stimulated with 1 μg/mL LPS for 0 h, 3 h, 6 h, 12 h, and 24 h. The expression levels of TNF-α, IL-6, IL-1β, and IL-12 were measured by qRT-PCR. GAPDH was used as a normalization control. The results are shown as the mean ± SD (*n* = 3). The *p* values were calculated using a two-sided Student’s *t*-test. * *p* < 0.05, ** *p* < 0.01, *** *p* < 0.001.

**Figure 3 ijms-20-01323-f003:**
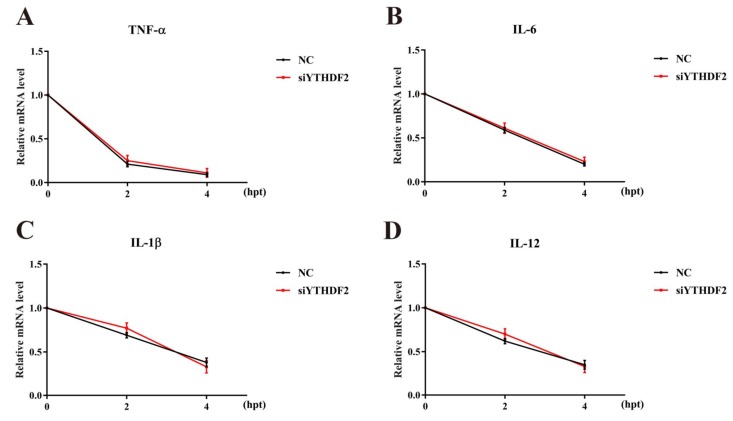
Effect of YTHDF2 knockdown on stability of TNF-α (A), IL-6 (B), IL-1β (C), and IL-12 (D) mRNAs. RAW 264.7 cells were transfected with YTHDF2 siRNA (siYTHDF2) or negative control siRNA (NC) for 24 h and stimulated with 1 μg/mL LPS for 6 h; then, 5 µg/mL actinomycin D was added to the cells to inhibit global mRNA transcription for 0 h, 2 h, and 4 h. The expression levels of TNF-α, IL-6, IL-1β, and IL-12 were measured by qRT-PCR. GAPDH was used as a normalization reference. The results are shown as the mean ± SD (*n* = 3). The *p* values were calculated using a two-sided Student’s *t*-test.

**Figure 4 ijms-20-01323-f004:**
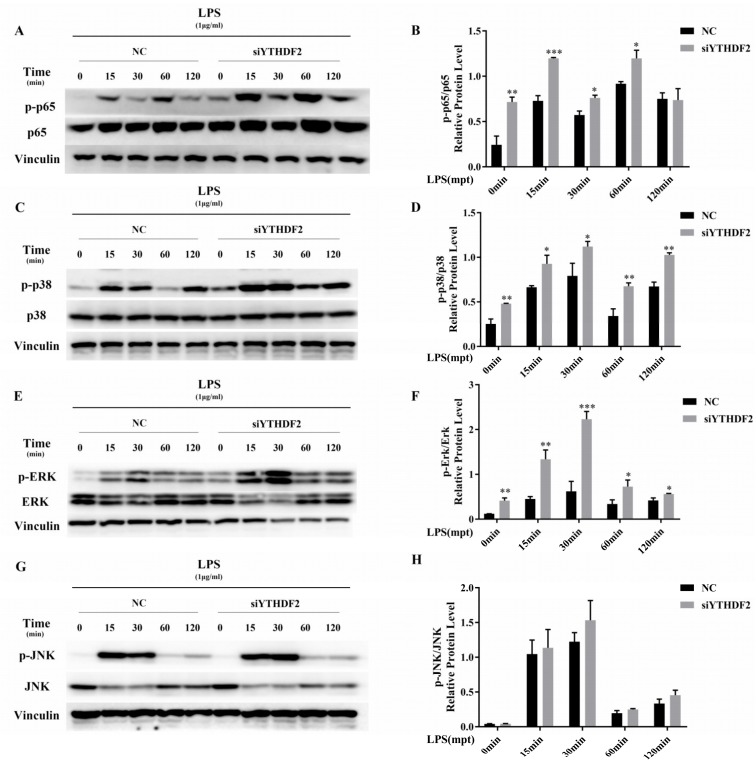
Effect of YTHDF2 knockdown on LPS-induced MAPK and NF-κB signaling. RAW 264.7 cells were transfected with YTHDF2 siRNA (siYTHDF2) or negative control siRNA (NC) for 24 h and then stimulated with 1 μg/mL LPS for 0 min, 15 min, 30 min, 60 min, and 120 min. (**A**–**H**) The phosphorylation levels of p65, p38, ERK, and JNK were measured by western blotting. Vinculin was used as an internal reference. p-p65/p65, p-p38/p38, p-ERK/ERK, and p-JNK/JNK values represent the relative level of signaling activation. The results are shown as the mean ± SD (*n* = 3). The *p* values were calculated using a two-sided Student’s *t*-test. * *p* < 0.05, ** *p* < 0.01, *** *p* < 0.001.

**Figure 5 ijms-20-01323-f005:**
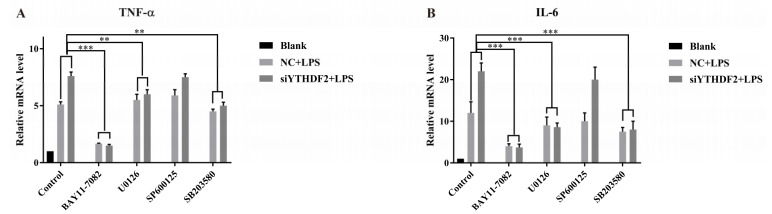
Effect of signaling pathway inhibitors on inflammatory cytokine expression in siYTHDF2-treated RAW 264.7 cells. RAW 264.7 cells transfected with YTHDF2 siRNA (siYTHDF2) or negative control siRNA (NC) were treated with the NF-κB pathway inhibitor BAY 11-7082, the p38 pathway inhibitor SB203580, the ERK pathway inhibitor U0126, or the JNK pathway inhibitor SP600125 for 1 h. Untreated cells were used as the blank control. The cells were then stimulated with 1 μg/mL LPS for 6 h. TNF-α (**A**) and IL-6 (**B**) expression levels were measured by qRT-PCR. GAPDH was used as a normalization reference. The results are shown as the mean ± SD (*n* = 3). The *p* values were calculated using a two-sided Student’s *t*-test. ** *p* < 0.01, *** *p* < 0.001.

**Figure 6 ijms-20-01323-f006:**
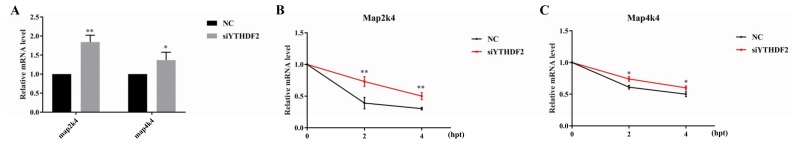
Effect of YTHDF2 knockdown on the expression and stability of MAP2K4 and MAP4K4 mRNAs. (**A**) RAW 264.7 cells were transfected with YTHDF2 siRNA (siYTHDF2) or negative control siRNA (NC) for 24 h and stimulated with 1 μg/mL LPS for 6 h. The MAP2K4 and MAP4K4 expression levels were measured by qRT-PCR. GAPDH was used as a normalization reference; (**B**,**C**) RAW 264.7 cells transfected with negative control siRNA (NC) or YTHDF2 siRNA (siYTHDF2) were stimulated with 1 μg/mL LPS for 6 h; then, 5 µg/mL actinomycin D was added to inhibit global mRNA transcription for 0 h, 2 h, and 4 h. MAP2K4 and MAP4K4 expression levels were measured by qRT-PCR. GAPDH was used as a normalization reference. The results are shown as the mean ± SD (*n* = 3). The *p* values were calculated using a two-sided Student’s *t*-test. * *p* < 0.05, ** *p* < 0.01.

**Table 1 ijms-20-01323-t001:** siYTHDF2 sequences for transcription (5′-3′).

siRNA	Sequences (5′-3′)
#1 siRNA	CCAUGAUUGAUGGACAGUCAGCUUU AAAGCUGACUGUCCAUCAAUCAUGG
#2 siRNA	GGGUGGAUGGUAAUGGAGUAGGACA UGUCCUACUCCAUUACCAUCCACCC
#3 siRNA	CCCAGUGGGAUUGACUUCUCAGCAU AUGCUGAGAAGUCAAUCCCACUCCC

**Table 2 ijms-20-01323-t002:** Primer sequences for qRT-PCR.

Gene	Forward Primer (5′-3′)	Reverse Primer (5′-3′)
*YTHDF2*	ATAGGAAAAGCCAATGGAGGG	CCAAAAGGTCAAGGAAACAAAG
*TNF-α*	CCACCACGCTCTTCTGTCTA	GGTCTGGGCCATAGAACTGA
*IL-1β*	CTTTGAAGTTGACGGACCCC	GCTTCTCCACAGCCACAATG
*IL-6*	CCTCTGGTCTTCTGGAGTACC	GGAGAGCATTGGAAATTGGGG
*IL-12*	GTGAACCTCACCTGTGACACGC	TGAATACTTCTCATAGTCCCTTTGG
*MAP2K4*	AATCGACAGCACGGTTTACTC	GCAGTGAAATCCCAGTGTTGTT
*MAP4K4*	CTGGCCGCCATCAAGGTTAT	AGCACCATAGTACGTGGCAAT
*GAPDH*	GCAAAGTGGAGATTGTTGCC	TGGAAGATGGTGATGGGCTT

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
