# Peer review of "m6A Reader YTHDF2 Regulates LPS-Induced Inflammatory Response"

_ijms, 2019, doi:10.3390/ijms20061323_

Round 1

Reviewer 1 Report

This study by Yu et al examines the role of the YTHDF2 RNA binding protein in the in the LPS-induced inflammatory reaction.  Using RAW 264.7 macrophages they demonstrate that YTHDF2 is induced at both the mRNA and protein level by LPS in a time dependent manner.  Depletion of YTHDF2 using RNAi revealed a role for YTHDF2 in suppressing LPS-induced IL-6, TNF-α, IL-1β and IL-12 expression and the phosphorylation of p65, p38 and ERK1/2 in NF-κB and MAPK signaling, perhaps indicating that YTHDF2 is part of a negative feedback loop.  The authors suggest that YTHDF2 controls NFkB and MAPK signaling by controlling the mRNA stability of the upstream proteins MAP2K4 and MAP4K4.

Overall the work is of interest to the field, the general increase in the inflammatory response in cells with depleted YTHDF2 is very intriguing.  It would however improve the paper if the authors showed a direct interaction between YTHDF2 and the MAP2K4 and MAP4K4 mRNAs by RNA IP.  As the authors state YTHDF2 can bind many RNA species and demonstrating this interaction in their model system would improve the paper.  Also, the statistical tests used for each analysis should be stated in the figure legends.

Author Response

1.     It would however improve the paper if the authors showed a direct interaction between YTHDF2 and the MAP2K4 and MAP4K4 mRNAs by RNA IP. As the authors state YTHDF2 can bind many RNA species and demonstrating this interaction in their model system would improve the paper.

We thank the reviewer for this valuable suggestion.

Previous studies have shown that MAP2K4 and MAP4K4 mRNAs are the target transcripts of YTHDF2 [1, 2]. The present study demonstrated that YTHDF2 knockdown stabilized the mRNA transcripts of MAP2K4 and MAP4K4, which consequently activated the MAPK and NF-κB signaling pathways and promoted the expression of proinflammatory cytokines. It is necessary to perform RNA IP to confirm the direct interaction between YTHDF2 and MAP2K4/MAP4K4 mRNA, as suggested by the reviewer. Although it is difficult to complete the experiment within a ten-day deadline of revision, we will conduct YTHDF2 RIP-sequence to verify the direct interaction between YTHDF2 and its target genes in a follow-up project.

2.     Also, the statistical tests used for each analysis should be stated in the figure legends.

We thank the reviewer for pointing it out. We have added the statistical tests used for each analysis to the figure legends. The revisions are highlighted in red in the manuscript.

References

1.          Li, Z.; Qian, P.; Shao, W.; Shi, H.; He, X. C.; Gogol, M.; Yu, Z.; Wang, Y.; Qi, M.; Zhu, Y.; Perry, J. M.; Zhang, K.; Tao, F.; Zhou, K.; Hu, D.; Han, Y.; Zhao, C.; Alexander, R.; Xu, H.; Chen, S.; Peak, A.; Hall, K.; Peterson, M.; Perera, A.; Haug, J. S.; Parmely, T.; Li, H.; Shen, B.; Zeitlinger, J.; He, C.; Li, L., Suppression of m(6)A reader Ythdf2 promotes hematopoietic stem cell expansion. Cell Res 2018.

2.          Wang, X.; Lu, Z.; Gomez, A.; Hon, G. C.; Yue, Y.; Han, D.; Fu, Y.; Parisien, M.; Dai, Q.; Jia, G.; Ren, B.; Pan, T.; He, C., N6-methyladenosine-dependent regulation of messenger RNA stability. Nature 2014, 505, (7481), 117-20.

Reviewer 2 Report

In the paper, authors investigate the potential function and molecular mechanism of RBP,YTHDF2 in the inflammatory response of mouse macrophages to LPS stimulation.

They have found that

1) LPS stimulation increases YTHDF2 expression.

2) YTHDF2 knockdown promotes LPS-induced inflammatory cytokine expression

3) YTHDF2 has a little effect on cytokine TNF-α, IL-1β, IL-6 and IL-12 mRNAs stability.

4) YTHDF2 downregulation activates LPS-induced NF-κB and MAPK signalling

5) Upstream kinases of NF-κB and MAPK signalling, MAP2K4 and MAP4K4 mRNA are targets of YTHDF2 that increases their stability.

It’s well-written paper containing experimental results that confirm an important role of the RBP,YTHDF2 in inflammation signalling

 Comments:

1. Authors should discuss their findings on the behaviour of expression YTHDF2 mRNA in Fig. 1 that shows its transient increase with the declining to the initial level while the inflammatory signal continues after 24 h (TNFa) in LPS stimulation.

2. Authors should discuss any molecular mechanisms of YTHDF2 mRNA upregulation in LPS-stimulated in macrophages. Are there any literature data or authors’ hypothesises on this mechanism?  Whether YTHDF2 mRNA upregulation is a result of NF-κB and MAPK signaling activation in LPS stimulation and it plays a positive feedback role in this signalling.

3. Authors should discuss the molecular mechanism of YTHDF2 binding to MAP2K4 and MAP4K4 mRNA.

4. Authors write “Recent studies have shown that MAP2K4 and MAP4K4 are the target genes of YTHDF2 [24, 32]”.  In this connection, authors should list all known gene targets of YTHDF2 and discuss those which are linked to inflammation. 

Author Response

Comments to the Author 

1.     Authors should discuss their findings on the behavior of expression YTHDF2 mRNA in Fig. 1 that shows its transient increase with the declining to the initial level while the inflammatory signal continues after 24 h (TNF-a) in LPS stimulation.

We thank the reviewer for this suggestion.

Previous studies have shown that YTHDF2 expression increases in multiple types of cells, such as pancreatic cancer, herpesvirus human cytomegalovirus (HCMV)-infected human foreskin fibroblasts, and heat shock-treated HepG2 cells [1-3]. Although these studies focused on the mechanism that YTHDF2 regulates its target genes, the molecular mechanism of YTHDF2 upregulation is not yet clear. YTHDF2 is an RNA bonding protein (RBP) that recruits Ccr4-Not1-deadenylation complex to destabilize mRNAs [4], which is similar to RBP Tristetraprolin (TTP) in functionality. TTP plays a significant role in regulation of the early inflammatory response by rapidly promoting the degradation of proinflammatory cytokine mRNAs [5]. TTP expression increases at the early stage of inflammation and then decrease after LPS stimulation for 1h, and the expression pattern of TTP is supposed to be related to its role in inflammatory response [6]. In the present study, the expression of YTHDF2 mRNA increased in response to LPS at the first 6 h and then decreased. We assumed that YTHDF2 might also participate in the regulation of early inflammation, which is similar to the role of TTP.

We have added relevant description to the discussion of the manuscript.

2.     Authors should discuss any molecular mechanisms of YTHDF2 mRNA upregulation in LPS-stimulated in macrophages. Are there any literature data or authors’ hypothesises on this mechanism?  Whether YTHDF2 mRNA upregulation is a result of NF-κB and MAPK signaling activation in LPS stimulation and it plays a positive feedback role in this signaling.

  As described above, YTHDF2 might participate in the regulation of early inflammatory reaction, which is similar to the role of TTP in inflammation. Previous researches showed that the transcription and activation of TTP rely on NF-κB and p38 signaling pathways [6, 7]. We assume that the upregulation of YTHDF2 expression might be also regulated by LPS-induced NF-κB and MAPK signaling activation, which needs further research.

3.     Authors should discuss the molecular mechanism of YTHDF2 binding to MAP2K4 and MAP4K4 mRNA.

We thank the reviewer for pointing it out. As an m6A reader, YTHDF2 targets the m6A modifications of RNA transcripts with its YTH domain. We used m6Avar (http://m6avar.renlab.org), a database of functional variants involved in m6A modification, to find the m6A modification of MAP2K4 and MAP4K4 mRNAs. The results showed that both MAP2K4 and MAP4K4 mRNAs have m6A modification sites. Previous studies have shown that MAP2K4 and MAP4K4 mRNAs are the target transcripts of YTHDF2 [8, 9]. Thus, it is reasonable to suppose that there is interaction between YTHDF2 and MAP2K4/MAP4K4 mRNA. We will conduct YTHDF2 RIP-sequence to verify the interaction between YTHDF2 and its target genes in a follow-up project.

4.     Authors write “Recent studies have shown that MAP2K4 and MAP4K4 are the target genes of YTHDF2 [24, 32]”.  In this connection, authors should list all known gene targets of YTHDF2 and discuss those which are linked to inflammation.

We thank the reviewer for this suggestion. More than three thousands YTHDF2 target genes were identified by PAR-CLIP and RIP-seq in recent studies [8, 9], among which we found MAP2K4 and MAP4K4 that can activate NF-κB and MAPK signaling pathways. Some other YTHDF2 target genes, such as IL-17RA, IL-17RB and IL-6R, may also play important roles in immune response, which need further research.

References

1.          Winkler, R.; Gillis, E.; Lasman, L.; Safra, M.; Geula, S.; Soyris, C.; Nachshon, A.; Tai-Schmiedel, J.; Friedman, N.; Le-Trilling, V. T. K.; Trilling, M.; Mandelboim, M.; Hanna, J. H.; Schwartz, S.; Stern-Ginossar, N., m(6)A modification controls the innate immune response to infection by targeting type I interferons. Nat Immunol 2019, 20, (2), 173-182.

2.          Yu, J.; Li, Y.; Wang, T.; Zhong, X., Modification of N6-methyladenosine RNA methylation on heat shock protein expression. PLoS One 2018, 13, (6), e0198604.

3.          Chen, J.; Sun, Y.; Xu, X.; Wang, D.; He, J.; Zhou, H.; Lu, Y.; Zeng, J.; Du, F.; Gong, A.; Xu, M., YTH domain family 2 orchestrates epithelial-mesenchymal transition/proliferation dichotomy in pancreatic cancer cells. Cell Cycle 2017, 16, (23), 2259-2271.

4.          Du, H.; Zhao, Y.; He, J.; Zhang, Y.; Xi, H.; Liu, M.; Ma, J.; Wu, L., YTHDF2 destabilizes m(6)A-containing RNA through direct recruitment of the CCR4-NOT deadenylase complex. Nat Commun 2016, 7, 12626.

5.          Astakhova, A. A.; Chistyakov, D. V.; Sergeeva, M. G.; Reiser, G., Regulation of the ARE-binding proteins, TTP (tristetraprolin) and HuR (human antigen R), in inflammatory response in astrocytes. Neurochem Int 2018, 118, 82-90.

6.          Tiedje, C.; Diaz-Munoz, M. D.; Trulley, P.; Ahlfors, H.; Laass, K.; Blackshear, P. J.; Turner, M.; Gaestel, M., The RNA-binding protein TTP is a global post-transcriptional regulator of feedback control in inflammation. Nucleic Acids Res 2016, 44, (15), 7418-40.

7.          Chrestensen, C. A.; Schroeder, M. J.; Shabanowitz, J.; Hunt, D. F.; Pelo, J. W.; Worthington, M. T.; Sturgill, T. W., MAPKAP kinase 2 phosphorylates tristetraprolin on in vivo sites including Ser178, a site required for 14-3-3 binding. J Biol Chem 2004, 279, (11), 10176-84.

8.          Li, Z.; Qian, P.; Shao, W.; Shi, H.; He, X. C.; Gogol, M.; Yu, Z.; Wang, Y.; Qi, M.; Zhu, Y.; Perry, J. M.; Zhang, K.; Tao, F.; Zhou, K.; Hu, D.; Han, Y.; Zhao, C.; Alexander, R.; Xu, H.; Chen, S.; Peak, A.; Hall, K.; Peterson, M.; Perera, A.; Haug, J. S.; Parmely, T.; Li, H.; Shen, B.; Zeitlinger, J.; He, C.; Li, L., Suppression of m(6)A reader Ythdf2 promotes hematopoietic stem cell expansion. Cell Res 2018.

9.          Wang, X.; Lu, Z.; Gomez, A.; Hon, G. C.; Yue, Y.; Han, D.; Fu, Y.; Parisien, M.; Dai, Q.; Jia, G.; Ren, B.; Pan, T.; He, C., N6-methyladenosine-dependent regulation of messenger RNA stability. Nature 2014, 505, (7481), 117-20.

Reviewer 3 Report

The authors present that YTHDF2 regulates the LPS-induced inflammatory response in the murine macrophage cell line RAW264.7.

The design of the experiments is good, knockdown efficiencies of siRNAs were tested properly, qpCR, Western Blot and stability assays support the conclusions.

However, I have two major concerns which have to be addressed:

1) Even cell lines of the same genotype or cell lines from the same tissue display heterogeneity. The whole paper is based on results derived from a single cell line. To obtain useful results and check whether result hold true in other macrophage cell lines, the authors should do the major experiments at least in one more cell line.

2) You show qPCR and stability assays for Map2k4 and Map4k4 in the last figure and claim that these genes could be relevant for the YTHFD2-induced inflammatory response. However, you do not provide evidence for that. You have to perform knockdown experiments for Map2k4 and Map4k4 followed by Western Blot or functional assays to validate your hypothesis.

Author Response

Comments to the Author

1.     Even cell lines of the same genotype or cell lines from the same tissue display heterogeneity. The whole paper is based on results derived from a single cell line. To obtain useful results and check whether result hold true in other macrophage cell lines, the authors should do the major experiments at least in one more cell line.

We thank the reviewer for this comment. We had intended to investigate the effect of YTHDF2 on LPS-induced inflammatory response in RAW 264.7 cells and mouse bone marrow-derived macrophages (BMMs). However, there were not enough mouse and BMMs available due to logistic problem. RAW264.7 cells were used as macrophage model to gain first insight into our hypothesis, BMMs will be used as macrophage cell line model for further study.

2.     You show qPCR and stability assays for Map2k4 and Map4k4 in the last figure and claim that these genes could be relevant for the YTHFD2-induced inflammatory response. However, you do not provide evidence for that. You have to perform knockdown experiments for Map2k4 and Map4k4 followed by Western Blot or functional assays to validate your hypothesis.

We thank the reviewer for this suggestion. The present study showed that YTHDF2 knockdown promoted the activation of NF-κB, p38 and ERK signaling and then triggered the expression of TNF-α, IL-1β, IL-6 and IL-12 in LPS-induced inflammatory response in RAW264.7 cells. To further investigate the mechanism of the effect of YTHDF2 knockdown on NF-κB, p38 and ERK signaling, we examined upstream components of the NF-κB and MAPK pathways and found that YTHDF2 knockdown increased the stability and expression of Map2k4 and Map4k4.

Previous studies have proved that Map2k4 and Map4k4 can activate NF-κB and MAPK signaling pathways as the upstream components of these pathways. Map2k4 activates NF-κB and MAPK signaling pathways in response to cellular stress [1-3]. Map4k4 can promote NF-κB localization and activation; Map4k4 knockdown suppresses the activation of ERK, p38, and JNK in MAPK signaling [4-7]. Besides, Map2k4 and Map4k4 mRNAs are the target transcripts of YTHDF2 [8, 9]. Thus, it is reasonable to suppose that YTHDF2 knockdown upregulated the phosphorylation of NF-κB and MAPK signaling components through stabilizing MAP2K4 and MAP4K4 mRNAs, thus promoting the expression of proinflammatory cytokines in LPS-stimulated RAW 264.7 cells. We will conduct YTHDF2 RIP-sequence to verify the interaction between YTHDF2 and its target genes in a follow-up project.

References

1.          Wang, S.; Yin, B.; Li, H.; Xiao, B.; Lu, K.; Feng, C.; He, J.; Li, C., MKK4 from Litopenaeus vannamei is a regulator of p38 MAPK kinase and involved in anti-bacterial response. Dev Comp Immunol 2018, 78, 61-70.

2.          Kundumani-Sridharan, V.; Subramani, J.; Das, K. C., Thioredoxin Activates MKK4-NFkappaB Pathway in a Redox-dependent Manner to Control Manganese Superoxide Dismutase Gene Expression in Endothelial Cells. J Biol Chem 2015, 290, (28), 17505-19.

3.          Raman, M.; Chen, W.; Cobb, M. H., Differential regulation and properties of MAPKs. Oncogene 2007, 26, (22), 3100-12.

4.          Gao, X.; Chen, G.; Gao, C.; Zhang, D. H.; Kuan, S. F.; Stabile, L. P.; Liu, G.; Hu, J., MAP4K4 is a novel MAPK/ERK pathway regulator required for lung adenocarcinoma maintenance. Mol Oncol 2017, 11, (6), 628-639.

5.          Virbasius, J. V.; Czech, M. P., Map4k4 Signaling Nodes in Metabolic and Cardiovascular Diseases. Trends Endocrinol Metab 2016, 27, (7), 484-492.

6.          Flach, R. J. R.; Skoura, A.; Matevossian, A.; Danai, L. V.; Zheng, W.; Cortes, C.; Bhattacharya, S. K.; Aouadi, M.; Hagan, N.; Yawe, J. C.; Vangala, P.; Menendez, L. G.; Cooper, M. P.; Fitzgibbons, T. P.; Buckbinder, L.; Czech, M. P., Endothelial protein kinase MAP4K4 promotes vascular inflammation and atherosclerosis. Nature Communications 2015, 6.

7.          Huang, H.; Tang, Q.; Chu, H.; Jiang, J.; Zhang, H.; Hao, W.; Wei, X., MAP4K4 deletion inhibits proliferation and activation of CD4(+) T cell and promotes T regulatory cell generation in vitro. Cell Immunol 2014, 289, (1-2), 15-20.

8.          Li, Z.; Qian, P.; Shao, W.; Shi, H.; He, X. C.; Gogol, M.; Yu, Z.; Wang, Y.; Qi, M.; Zhu, Y.; Perry, J. M.; Zhang, K.; Tao, F.; Zhou, K.; Hu, D.; Han, Y.; Zhao, C.; Alexander, R.; Xu, H.; Chen, S.; Peak, A.; Hall, K.; Peterson, M.; Perera, A.; Haug, J. S.; Parmely, T.; Li, H.; Shen, B.; Zeitlinger, J.; He, C.; Li, L., Suppression of m(6)A reader Ythdf2 promotes hematopoietic stem cell expansion. Cell Res 2018.

9.          Wang, X.; Lu, Z.; Gomez, A.; Hon, G. C.; Yue, Y.; Han, D.; Fu, Y.; Parisien, M.; Dai, Q.; Jia, G.; Ren, B.; Pan, T.; He, C., N6-methyladenosine-dependent regulation of messenger RNA stability. Nature 2014, 505, (7481), 117-20.

Round 2

Reviewer 2 Report

I am satisfied with the authors' responses to my comments.

Reviewer 3 Report

I am okay with the revised submission.